# Barriers to evidence-based acute stroke care in Ghana: a qualitative study on the perspectives of stroke care professionals

Leonard Baatiema,[1,2] Ama de-Graft Aikins,[1] Adem Sav,[3] George Mnatzaganian,[4] Carina K Y Chan,[5] Shawn Somerset[3]

► Prepublication history and additional material is available online. To view please visit the journal online (http://dx.doi.org/ 10.1136/bmjopen-2016-015385)

[1]Regional Institute for Population Studies, University of Ghana, Accra, Legon, Ghana
[2]School of Allied Health, Faculty of Health Sciences, Australian Catholic University, Sydney, Australia
[3]School of Allied Health, Faculty of Health Sciences, Australian Catholic University, Brisbane, Australia
[4]College of Science, Health and Engineering, La Trobe Rural Health School, La Trobe University, Melbourne, Australia
[5]School of Psychology, Faculty of Health Sciences, Australian Catholic University, Brisbane, Australia

**Correspondence to**
Leonard Baatiema;
baatiemaleonard@gmail.com

## ABSTRACT

**Objective** Despite major advances in research on acute stroke care interventions, relatively few stroke patients benefit from evidence-based care due to multiple barriers. Yet current evidence of such barriers is predominantly from high-income countries. This study seeks to understand stroke care professionals' views on the barriers which hinder the provision of optimal acute stroke care in Ghanaian hospital settings.
**Design** A qualitative approach using semistructured interviews. Both thematic and grounded theory approaches were used to analyse and interpret the data through a synthesis of preidentified and emergent themes.
**Setting** A multisite study, conducted in six major referral acute hospital settings (three teaching and three non-teaching regional hospitals) in Ghana.
**Participants** A total of 40 participants comprising neurologists, emergency physician specialists, non-specialist medical doctors, nurses, physiotherapists, clinical psychologists and a dietitian.
**Results** Four key barriers and 12 subthemes of barriers were identified. These include barriers at the patient (financial constraints, delays, sociocultural or religious practices, discharge against medical advice, denial of stroke), health system (inadequate medical facilities, lack of stroke care protocol, limited staff numbers, inadequate staff development opportunities), health professionals (poor collaboration, limited knowledge of stroke care interventions) and broader national health policy (lack of political will) levels. Perceived barriers varied across health professional disciplines and hospitals.
**Conclusion** Barriers from low/middle-income countries differ substantially from those in high-income countries. For evidence-based acute stroke care in low/middle-income countries such as Ghana, health policy-makers and hospital managers need to consider the contrasts and uniqueness in these barriers in designing quality improvement interventions to optimise patient outcomes.

## BACKGROUND

Recent significant technological advancement in medical practice has increased demands, expectations and pressures on healthcare staff to provide quality and evidence-based care. This is exacerbated by the wide knowledge-clinical practice gap across the world,[1] particularly in low/

## Strengths and limitations of this study

► This study represents the first in Ghana to explore, in-depth, the barriers perceived by stroke care professionals to optimum provision of acute care in hospital settings.
► The work focused exclusively on the perspectives of acute stroke care professionals from diverse professional disciplines, expertise, gender, tertiary and non-tertiary hospitals across different geographical settings.
► This study did not focus on barriers to a specific stroke care intervention, as reported extensively in previous works, but rather on barriers across the continuum of stroke care.
► The study reported results from a limited set of participants whose views may not be reflective of the wider health staff responsible for acute stroke care in Ghana.
► Given the qualitative nature of the study, data interpretation could be subjective and thus, caution should be applied in interpretation.

middle-income countries[2] where research translation has become an urgent healthcare agenda.[3] Empirical evidence in the USA and Europe, for example, demonstrates how only about 30% to 50% of patients receive evidence-based interventions in clinical settings.[4 5] It is further suggested that translation of an evidenced-based health intervention into routine clinical practice can take up to 17 years.[6] The need to identify barriers that underpin the slow uptake of evidence-based care in clinical settings is essential in understanding the extent to which health professionals provide such care to patients.[5 7] As a result, theoretical and conceptual attempts have been made to shed light on the factors which affect the current knowledge-practice gap in healthcare settings.[5 8 9]

Due to the increasing global stroke-related mortality and morbidity,[10] the past decades have witnessed a

proliferation of evidence-based acute stroke care interventions.[11–16]Throughout this paper, the term evidence-based acute stroke care interventions also referred to as optimal acute stroke care comprised all acute stroke care interventions based on scientific evidence, clinical judgement and expertise of a clinician and the needs of patients.[17] Other key stroke experts have also recommended essential components of an evidence-based acute stroke care for improved patient outcomes.[18 19] Notwithstanding such advances, uptake of such recommendations in clinical settings remains slow,[20 21] suggesting that only a small proportion of stroke patients receive optimal care. Although the low uptake of these interventions is a global health challenge, evidence suggests the pace of uptake in high-income countries exceed that of low/middle-income countries.[20–22] Numerous barriers have been identified to explain the low uptake of such evidence-based stroke care interventions into routine clinical practice. Some of these barriers include inadequate medical facilities, inadequate knowledge and skill levels of stroke care providers, low awareness of current acute stroke care interventions and the perceived efficacy levels of acute stroke care interventions.[23–25] There are also barriers at the patient level which include delays in seeking emergency care due to lack of awareness of early stroke symptoms or financial constraints.[26 27]

Although research has increased our knowledge about the range of barriers to the uptake of evidence-based stroke care in clinical settings, a more balanced and holistic understanding of such research is needed. Existing research to date only presents a one-sided view, and bias towards high-income countries (eg, Australia and USA) and moreover, is focused primarily on barriers inhibiting uptake of thrombolytic therapy.[23 25 27 28] However, only few studies have looked at barriers to other components of acute stroke care interventions.[24 29 30] A study by Langhorne and colleagues also provides insightful information on the uptake of stroke unit care components in resource-poor settings.[20] It is unclear if these barriers apply to low/middle-income countries such as Ghana, where the geopolitical, socioeconomic and health system contexts vary. An investigation of such barriers is important in low/middle-income countries because the global stroke burden is much higher there,[10 31] and yet evidence suggests uptake of evidence-based acute stroke care interventions is relatively lower.[20 21]

This study therefore aimed to identify the views of stroke care professionals on barriers inhibiting the provision of optimal acute stroke care in Ghanaian hospitals, since such information is non-existent. Acute stroke care in this context applies to the provision of care in the initial days and weeks after a stroke. Greater insights about these barriers and how they differ according to hospital settings and across stroke care professional disciplines are important for developing interventions towards enhancing optimal patient outcomes in Ghana. The findings may also have broader relevance to other resource-poor settings.

## METHODS
### Study design
This study is part of a larger multisite study to evaluate the provision of evidence-based acute stroke care in acute care in major referral hospitals in Ghana. A qualitative study design using semistructured interviews was employed to gain a rich and in-depth understanding of the barriers faced by stroke care professionals. The importance of qualitative data to successful translation of best scientific evidence into clinical practice has also been recommended.[32] The study design, data collection, analysis and reporting were conducted in accordance with the consolidated criteria for reporting qualitative research[33] as shown in online supplementary file 1.

### Settings
The study was conducted in a convenient sample of three referral tertiary (teaching) and three regional (non-teaching) hospitals from the southern, middle and northern belts of Ghana, between November 2015 and April 2016. This represents three of the five tertiary-teaching hospitals and three of the nine regional hospitals in Ghana. The study hospitals are major referral hospitals for other hospitals and health centres located in 6 of the 10 administrative regions of Ghana and were chosen to account for the geographical and socioeconomic contrasts among the 10 administrative regions of the country. The hospital bed capacity for these hospitals is as low as 150 for the regional hospitals, whereas the teaching hospitals bed capacity is approximately 653. The tertiary hospitals are larger referral centres and are well resourced with diagnostic and therapeutic facilities, while the regional hospitals act as major referral points to other hospitals and health centres within their catchment areas. Overall, the annual stroke admissions for 2014 ranged from 49 for the regional and 1500 stroke cases for the teaching hospitals. See online supplementary file 2 for additional information on the study hospitals.

### Research team
Participants have no prior relationship with the researchers but because of the previous works of two of the researchers (LB and Ad-GA) in some of the study regions, it is possible participants have met or are aware of their works. LB is a health services researcher with interest in health services and policy research, research on implementation science and quality improvement interventions for stroke care health professionals. He is skilled in both qualitative and quantitative research works. Ad-GA conducts social and health psychology research using largely qualitative methods. AS is a health services researcher employing a mixed methods approach. The remaining researchers (GM, CKYC and SS) on the other hand, also have relevant skills, knowledge and interest in qualitative studies and the topic under study. . Overall, the research team comprised two women and four men.

**Table 1** Distribution of interview participants and study hospitals

| Participants | Tertiary/teaching hospitals | Regional hospitals | Total |
|---|---|---|---|
| Nurses | 11 | 9 | 20 |
| Medical doctors/physicians | 6 | 6 | 12 |
| Clinical psychologist | 1 | 1 | 2 |
| Physiotherapist | 2 | 3 | 5 |
| Dietitian | 0 | 1 | 1 |
| Total | 20 | 20 | 40 |

## Participants

Participants comprised key hospital staff, primarily involved in directing or providing acute care for stroke patients. To achieve maximum variation in the continuum of care that would reflect a real life setting, the study recruited nurses, specialist medical doctors (neurologists, emergency physician specialist), non-specialist medical doctors, clinical psychologists, physiotherapists and a dietitian, representing diverse expertise and experience relevant to acute stroke care. Table 1 shows participants' distribution across study sites.

## Sampling and recruitment

Purposive sampling was used to recruit all study participants. Participant recruitment was facilitated by two of the researchers (LB, Ad-GA). . To commence recruitment and promote the study to eligible participants, meetings were held with hospital administrators, in-service training and research coordinators, department heads and nurses incharge in the study hospitals. Potential participants were then recommended from these meetings and engagements. Initial contact with prospective participants was made face-to-face or by telephone calls by the first author to identify the date, time and venue for the interviews. Potential participants were identified. Due to time and workload restrictions, three participants declined to participate in the study. The number of participants enrolled into the study was determined by data saturation.

## Data collection

All interviews were conducted face-to-face in English by LB. Data collection was conducted in various venues including: general and emergency wards, consulting rooms, conference rooms, participants' office rooms and physiotherapy departments. The interviews were facilitated by an interview guide (see online supplementary file 3) developed by the researchers and informed by an extensive literature review on the topic. The interview guide was pilot-tested with three nurses and three medical doctors at non-study sites and adapted to reflect the professional role of the interviewees. With the permission of interviewees, each interview was recorded using a digital voice recorder. Detailed field notes were also taken. The study repeatedly used prompts to facilitate the elicitation of more and clearer information or clarification

of certain concepts used by participants. The interviews lasted 45 min on average and all recorded interviews were transcribed verbatim by professional transcribers for the final data analysis. About a third of the transcripts were shared with selected participants to crosscheck and ensure the information reflected the interview process and 13 transcripts were returned.

## Data analysis

Thematic data analysis,[34][35] combined with some elements of a grounded theory approach,[36] were used to analyse the data. Pre-existing thematic categories based on relevant literature were used in the data analysis. The grounded theory took an inductive approach to ensure all essential emergent themes from the codes not included in the deductive pre-existing coded list of barriers were captured. An initial codebook based on prior codes was developed and subsequently modified with the addition of new emergent themes after a line-by-line reading and rereading of transcripts by one author (LB). A second author (Ad-GA), crosschecked the final coded results with a sample of the transcripts. Using the constant comparison approach,[36][37] a comparative analysis of both emergent and prior themes was conducted between study sites and participants to understand areas of convergence and divergence. NVivo software package V.10.0 (38)[38] was employed to organise, code and identify all data.

Trustworthiness and transferability in the study results were facilitated by the consistent use of the interview guide during the interview process, audio recording of all interviews, professional transcription of the interviews and the use of the NVivo software to manage the entire data analysis process. As a measure to further enhance data trustworthiness,[39] some transcripts were shared with selected participants for crosschecking, known as member validation.

## FINDINGS

A total of 40 participants took part in the study, approximately 6 participants per study site. Participants included both men and women of varied professional disciplines, ranks and years of practice in the study sites (see table 1).

**Table 2** Themes and definitions

| Coding categories | Definition of barriers |
|---|---|
| Patient level | Includes factors, such as late arrival or low awareness of stroke symptoms, denial of stroke, financial capacity, sociocultural practices or beliefs inhibiting access or adherence to optimal acute stroke care. |
| Hospital or health system level | Relates to a lack of inadequate medical facilities or equipment, staff numbers, protocols, management support, supporting policies, organisational context or norms which support implementation of standard care and availability of staff professional development opportunities to support the provision of standard care. |
| Stroke care professionals | Describes acute stroke care providers' level of team support, communication or collaborations which affect the provision of care. Also includes competence, skill, knowledge, awareness, familiarity or agreement to specific treatments, their values, motivations or attitudes towards particular treatments or intervention. |
| National/state health policy context | Relates to the level of political will for acute stroke care in the form of national stroke policies, limited allocation of resources for acute stroke care, reimbursement of funds to hospitals, national health policies to support stroke patients' access to optimal care and the lack of any regulatory frameworks or policies to support stroke care. |

## Barriers to acute stroke care

Four key themes of barriers to the provision of optimal acute stroke care emerged from the data; patient, hospital or health system, healthcare providers and national health policy factors. Table 2 describes each of these barriers. Embedded in these themes were 12 subthemes which provided specific and contextualised meaning to the main themes.

### Patient factors

Under this category of barriers, five subthemes were identified: financial constraints, delays, sociocultural or religious beliefs and practices, discharge against medical advice and denial of stroke.

### Financial constraints

In all the study sites and across participants from the various professional disciplines, barriers such as lack of funds to transport patients to the hospital, inability to pay for medical expenses (eg, CT brain scanning services, laboratory tests and other healthcare associated expenses) were consistently raised. Patients' or caregivers' decision to first seek medical care, organise means of transport to the hospital or pay for medical expenses were often constrained by their level of financial capacity. As a result, access to care was often delayed or deprived. An excerpt from a participant emphasised this:

'poverty and ability to pay for medical cost is the issue over here,…let's say a doctor will request a patient to do a CT scan, do some lab tests …, but the patient just simply cannot afford it, or it takes too long for them to gather the money, so for two, three and sometimes

five weeks you are treating a patient without a CT scan investigation' (Medical doctor, ID 9)

### Delays

Patients' late arrival to the hospital was commonly cited as another barrier to acute stroke care. Participants suggested the reasons for such delays arose from their lack of awareness of early stroke symptoms and decision to first seek herbal or faith-based, rather than medical care. Delays were also attributed to financial capacity of the family to seek medical care, especially in instances where the family breadwinner was the stroke victim. Hence, patients with good financial circumstances were more likely to seek early acute medical care compared with those with poor finances:

'…they don't bring the patients early and when they come, they will tell you the condition just started, that they just noticed the symptoms and rushed the patient to the hospital. But you realize that this patient had the stroke for long, not very acute as they described, either they have sought treatments elsewhere or other interventions before arriving here' (Nurse, ID 4)

On the low awareness of early stroke symptoms, one participant noted:

'They don't have knowledge of early stroke symptoms, they are ignorant about stroke symptoms…, Because they don't know what the condition is, patients or families will rather prefer to self-medicate with painkiller or remain at home upon symptoms onset

with the hope that the symptoms will disappear' (Medical doctor, ID2)

## Sociocultural and religious beliefs

Patients' sociocultural or religious beliefs and practices emerged as another predominant barrier. Cultural beliefs and practices (eg, view stroke having a spiritual cause, retribution from their gods and not a condition which can be managed medically) were often very important and likely to influence patient health-seeking behaviour. A common practice from such beliefs was patients' desperate attempts to defer medical care for herbal or traditional medical care or make attempts to combine both while hospitalised. Some nurses noted such practices or beliefs have often compelled families of patients to abandon medical care in the hospital for alternative care provided by traditional or faith healers. For example, all physiotherapists interviewed believed sociocultural beliefs and practices have limited patient attendance of outpatient care after discharge as most resorted to local herbal treatment options or to prayer camps. A participant had this to say about the religious beliefs on treatment compliance:

'…they become very spiritual once they are diagnosed with a stroke; most now want to focus on their spiritual life instead. You realize that consistently our stroke patients want to talk about God, talking about how lucky they have been, how God has saved them from death' (Clinical psychologist, ID 1)

## Discharge against medical advice

'Discharge against medical advice' was consistently discussed by participants and emerged as a key barrier to optimal acute stroke care. This practice was generally perpetuated by two factors; financial capacity to meet medical expenses and families' desire to resort to other forms of care such as traditional herbal medicine, consultation of spiritualists or faith healers. Participants attributed the increased patient and family interest to such alternative forms of care to the vibrant advertisements across the media by traditional herbal medicine practitioners and faith healers. Indeed, promises were made by such individuals to cure stroke and other chronic conditions within a week or two after commencing treatment:

'I recently heard one advertisement which said acute stroke patients should just come here and will be made to walk within a week. So it has gotten to the point where patients easily get misled by these adverts, they find these traditional or faith healers attractive and accessible' (Medical doctor, ID 11)

According to nurses, such incidences were also linked to the sociocultural beliefs and practices of the people where health conditions such as stroke, were associated with supernatural or spiritual causes. After being informed about their stroke condition, some stroke patients often

insisted on being discharged. Moreover, refusal to heed patients' or families' requests for early discharge often resulted in non-compliance to treatment, sudden disappearance of patients or desertion of patients by family members:

'with the relatives, as soon as they find out that it is a stroke, they start finding ways of transferring the patient to seek herbal medication or to a prayer camp …, so they request for discharge against medical advice and take the patient away' (Nurse, ID 15)

## Denial

It was also reported that some family members or patients sometimes rejected the diagnosis and dissociated their condition from stroke after being informed about the condition. Participants even acknowledged instances where some family members challenged their professional competence because they felt an incorrect diagnosis was made. The denial of stroke stemmed from the diverse misunderstandings of the illness, with some patients/family members viewing it as an attack or retribution from their gods or spirits for a wrongdoing. In such situations, the provision of care was difficult, as some family members were less compliant during treatment:

'I remember one care giver following up to me to inquire whether we were sure the condition of their relative was a stroke, as she believed a wrong diagnosis was documented. Because to her, their relative does not deserve to have a stroke' (Nurse, ID 11)

## Hospital or health system factors

The subthemes of these system factors were shortage of medical facilities/equipment, lack of a stroke specific protocol, inadequate staff numbers and limited staff professional development opportunities.

## Shortage of medical facilities

The limited availability of essential medical equipment to facilitate effective provision of acute stroke care was a common feature in study hospitals within the northern belt. There was a shortage of medical facilities such as blood pressure (BP) monitoring apparatus, cardio monitors, suction machines, adjustable hospital beds and inadequate space to facilitate patient care. For example, participants in the only stroke unit in this study believed that the inadequate bed capacity (six-bed capacity) limited admission of many patients to receive optimal care. One participant commented:

'Unfortunately, you find stroke patients, they come in, no bed, they are sitting on chairs, sitting on the wheel chair or on the bare floors, these are the conditions under which we are expected to provide standard care…"(Medical doctor, ID 2)

This experience and another comment below exemplified this barrier:

> 'We have just one oxygen for all the patients in this ward so the nurses are sometimes compelled to use their discretion to wean patients off oxygen to enable another patient benefit if his/her condition is more severe' (Medical doctor, ID 7)

Additionally, the lack of a stroke unit was a common concern expressed by medical doctors from hospitals in the middle and southern belts, a situation they believed was caused by limited funds allocated by hospitals and a low priority for acute stroke care. A lack of medical equipment and consumables could delay or deprive patients of standard care. Participants also talked about instances where some medical doctors acquired personal BP monitoring devices to support patient care because of shortages. Another issue was the absence or frequent malfunction or breakdown of diagnostic services such as CT scanning services, a situation which often delayed care delivery or led to referral of patients to other hospitals. According to some medical doctors, this situation sometimes compelled them to proceed with care delivery without a CT scan investigation to inform treatment options:

> 'I can say the biggest problem we face is our diagnostic equipment. See the whole of this so called big hospital, we have only one CT scan machine. The machine has been out of service for over 6 to 8 weeks and was only put to use again two weeks ago …,' (Medical doctor, ID 11)

### Lack of a specific protocol for acute stroke care
Most nurses believed the absence of a specific protocol or clinical guideline for acute stroke care was a key barrier:

> '….sometimes the cases come and you've forgotten some important procedures because I left the classroom a very long time ago' (Nurse, ID 13)

One nurse recounted her experience of providing an acute stroke care with much uncertainty because there was no medical doctor or a senior colleague to guide her. This nurse stressed the importance of a clinical protocol, which she argued could facilitate the provision of standardised care even in the absence of a specialist or a medical doctor:

> 'Most of the stroke cases I have witnessed were rushed in here during late hours, sometimes after mid-night and most times, its only nurses present to attend to the case. So the patient has to wait until a doctor arrives, sometimes the next morning and that is why I think the protocol will at least guide us to safely initiate initial treatment' (Nurse, ID 18)

### Limited staff
Limited staff especially stroke specialists (eg, neurologists, neurosurgeons and trained stroke nurses) were also a key barrier across the study sites. This issue was more dominant in the non-tertiary regional hospitals and participants in the northern belt of Ghana. Participants, especially nurses, believed the current staff numbers were inadequate to provide optimal acute stroke care (eg, regular checking of BP levels, sugar levels, regular turning of patients to prevent pressure sores and management of urinal incontinence to minimise risk of urinary tract infections).They expressed frustration about the high workload, which often compromised effective patient care:

> 'you could have patients running over 40,…, some are in the wheel chair, some are on the chairs you see over there, some are on the beds, sometimes some are on the stretchers" (Nurse, ID 5)

### Limited staff professional development opportunities
With the exception of medical doctors, nurses and allied health staff expressed great interest in opportunities for staff professional development, mainly in hands-on training workshops related to stroke clinical care. Although there were policies to support staff develop their current knowledge and skills, such opportunities were very rare. Nurses, for example, emphasised the importance of continuous education and professional development as current clinical practice was underpinned by what they were taught in schools many years ago. Overall, there was strong opinion on this matter and a lack of continuous training opportunities inherently affected the quality of care provided to acute stroke patients:

> 'we don't have regular workshops…, even if there will ever be such an opportunity, you will only consider attending provided you can afford the cost as this hospital won't support us attend such a workshop' (Nurse, ID 4)

### Healthcare providers' factors
Two main subthemes of barriers were identified at the healthcare staff level; limited knowledge in acute stroke care and inadequate team collaboration and coordination.

### Inadequate knowledge
Lack of knowledge on how to provide appropriate treatment was often discussed, particularly by nurses. Unlike the medical doctors, the nurses were unaware of thrombolytic therapy. This particular type of therapy was not part of what the medical doctors recommended for acute ischaemic stroke care:

> 'What did you say again? thromboly… what? Not here, I am hearing thrombolysis for the first time. It is not part of our treatment plan for stroke patients in this hospital. How come you say it is one of the

key therapies for acute ischemic stroke and I am not aware of it?' (Nurse, ID 9)

Most nurses also identified insufficient knowledge of certain acute stroke care procedures as a barrier, especially in triaging unconscious stroke patients. They expressed uncertainty about their ability to often proceed with care delivery in the absence of a medical doctor. Although nurses talked about consulting senior colleagues, some level of uncertainty was still noted in proceeding to provide care in the absence of a medical doctor. Despite their clinical training, nurses cited difficulties arising from efforts to respond to unconscious stroke patients and conduct assessment to support an accurate stroke prognosis:

'Sometimes a stroke case arrives unconscious and you start shivering especially when it is in the night and there is no doctor around to respond immediately. I feel very nervous when I realise I am the only senior nurse in the ward to attend to this patient' (Nurse, ID 12)

### Team collaboration and communication

According to most nurses and all allied health staff, collaborative work in a multidisciplinary stroke team was inadequate, and an obstacle to effective patient care. Physician driven stroke care without adequate involvement of other staff, was frequently discussed:

'I don't even think we have a working team here, it is more of a doctor giving instructions…,giving instructions to nurses, though nurses are there with the patients 24 hours, the medical doctors just come to see their patients and then disappear. Is that what you call teamwork?' (Nurse, ID 14)

Allied health staff expressed a sense of marginalisation and disconnectedness, especially in the early stages of care. A dietitian for example cited instances where medical teams (doctors and nurses) often discharge patients without his view on dietary plans at discharge. Three physiotherapists expressed similar concerns of limited involvement which in their view inhibited their ability to develop initial rapport with patients or the opportunity to educate patients about the importance of self-care practices, following discharge:

'There was an occasion my rounds coincided with the medical team's rounds; I quickly joined them. I made a suggestion on a particular patient we were attending to but this was brushed off and the medical doctor behaved as if I was trying to direct him what to do or take over his job' (Physiotherapist, ID 3)

### National policy context factors

Participants identified one key barrier under this theme; lack of political will for acute stroke care.

### Lack of political imperative

The lack of national level support and political imperative for acute stroke care was consistently cited as a broad level barrier, particularly by medical doctors. They expressed strong views on this issue, attributing it to the increasing out-of-pocket medical expenses for patients. Despite the existence of the national health insurance policy which was supposed to replace the practice of 'cash and carry', a lack of political imperative for the scheme has gradually introduced the policy of upfront payments by patients prior to acute care in most hospitals in Ghana presently. They believed this has negatively affected patients' access to care (not only stroke patients) because of their inability to pay for medical expenses. The limited coverage of the national health insurance scheme on chronic care, such as stroke, was also stated as a key barrier. Patients experienced difficulties paying for stroke-related medical costs (eg, CT brain scans and other laboratory tests) that were not covered by the national health insurance scheme. Overall, there was a sense of powerlessness about national level neglect for acute stroke care. Consequently, this resulted in staff dissatisfaction and a lack of motivation to provide effective care:

'The problems we face in our current health sector has very little to do with health professionals' reluctance to provide standard care. It is the health system!…, we are under a system where every medication is expensive for the ordinary Ghanaian to afford and yet, we make the patients to believe the national health insurance policy covers everything. Now almost every medication has to be paid for by the patient and if they can't afford what we recommend as best treatment option for their condition, we provide the alternatives which may not be very effective' (Medical doctor, ID 11).

## DISCUSSION
### Summary of main findings

This study provides in-depth insights of the barriers to the delivery of optimal acute stroke care in Ghana, a largely neglected low/middle-income country in Africa. The findings suggest that although the barriers identified share some commonality with those reported in previous studies in high-income countries, some barriers are unique to optimal stroke care in low/middle-income settings such as Ghana. Some of the predominant barriers to acute stroke care in high-income countries often comprised patient delay in seeking early care, inadequate medical facilities to support optimal patient care, healthcare providers' attitudes towards some acute stroke care interventions, poor communication and lack of cooperation among healthcare providers.[23–25] On the contrary, although there is an overlap of these barriers in both high-income and low/middle-income countries, the issue of discharge against medical advice

and the role of sociocultural and religious beliefs or practices of stroke patients and their families characterise the present study. While illuminating the reasons influencing the provision of optimal acute stroke care in a typical resource-poor setting, the findings further unravel barriers peculiar to different stroke care professionals and hospital settings where much policy attention is required for effective and timely translation of evidence-based stroke care intervention into routine clinical practice.

## Comparison with previous literature

As found in this study, stroke care in high-income countries with modern resources has consistently reported barriers corresponding to patient, health system/hospital, health staff and the national level factors.[23 25 40] Highlighting patient level barriers as the most predominant of all barriers identified in our study, a previous study corroborated this by reporting 91% of participants viewed prehospital delay at the patient level as the most dominant barrier to providing thrombolytic therapy.[27] An earlier study in Ghana[41] found that only 40% (277/693) could correctly identify stroke symptoms, reinforcing the importance of our finding that participants identified patient delays to seek care due to low awareness of stroke symptoms.

Although our findings on the importance of patient level barriers to optimal stroke care are in line with previous research,[23 27 30 42] the explanation and the circumstances in which some patient level factors acted as barriers to optimal acute stroke care were somewhat different . For example, sociocultural or religious beliefs and practices were perceived to underpin health-seeking behaviours of stroke patients and their families in Ghana. Although this is inconsistent with the literature on barriers to acute stroke care in high-income countries, our findings corroborate with research on other chronic diseases and health-seeking behaviour in Ghana.[43 44]This underscores the influence of such beliefs and practices to health-seeking behaviours of patients and families in Ghana. Evidence within the African contexts suggests patient access to traditional and faith healers as complimentary avenues of care is due to the easy access, lower cost and cultural legitimacy of such alternatives.[45 46]

In addition, patient discharge against medical advice was also a key barrier affecting optimal clinical care. This finding is also largely inconsistent with published barriers to acute stroke care from high-income countries. Despite the limited popularity of such barriers in previous studies, this has been well articulated in other health contexts and conditions with conclusive arguments of the practice being a drawback and an obstacle to provision of adequate and quality healthcare.[47–50] Clearly, this issue requires further investigation in Ghana, and possibly other low/middle-income countries. L

Other important barriers from this study were related to the health system, such as limited stroke care specialists, increased workload for staff, inadequate medical facilities,

lack of protocols and unavailability or limited access to CT brain scans. The importance of this set of barriers has been reported previously,[23 29 51] highlighting the extent to which they affect provision of optimal patient care. For example, one study found that 71% of participants identified lack of protocols, care paths and opportunities for staff professional education as important barriers to the provision of optimal acute stroke care.[27] Comparable to our study, a Swedish study identified low staffing levels as a major barrier to optimal stroke care.[25] Despite these studies being conducted in high-income countries, their corroboration with the present study reinforces the importance of hospital/health system level barriers to the uptake of evidence-based practice.

The issue of limited collaboration or involvement of allied health staff and other providers in the provision of care is also worthy of attention. Multidisciplinary and coordinated care remains a central component in contemporary evidence-based practice for acute stroke care.[18 52] As a result, inadequate involvement of these staff is a significant issue since participants noted that their limited involvement is detrimental to optimal patient care. Evidence from existing scholarship on such barriers has been previously reported,[53–55] thus stressing the need to consider interventions to improve collaboration among stroke care professionals in acute stroke care.

Nurses' knowledge of acute stroke care interventions such as thrombolytic therapy was also identified as a barrier, consistent with previous studies.[23 27 40] This issue has also been identified in an Australian study where 50% of nurses reported having limited knowledge of thrombolytic therapy.[27] Such findings highlight the importance of this issue to optimal stroke care in both low/middle-income and high-income countries.

Finally, another barrier identified in the present study relates to the low level of political will for optimal acute stroke care. This barrier is evident in the absence of a national stroke clinical guideline, a national framework for quality improvement interventions for stroke and limited coverage of the national health insurance scheme to cover patients' medical expenses. While this finding corroborates with previous studies asserting the limited prioritisation of acute stroke care by health policy-makers in resource-poor settings,[56–59] this could also likely be symptomatic of the current limited global health funding for stroke and other non-communicable diseases compared with communicable diseases.[60]

## Implications for future research, policy and clinical practice

The present findings have several important implications for the provision of evidence-based acute stroke care in Ghana. First, patient financial constraints appear to be a key barrier to optimal care and needs urgent attention. It is apparent that patients and family members struggle with the financial costs of stroke treatment and strategies are needed to overcome this burden. The current Ghana National Health Insurance Policy offers limited financial risk protection for stroke care. This epitomises

the fragile nature of healthcare systems in low/middle-income countries, which require significant structural and policy reforms to minimise the current high cost of treatment for chronic diseases such as stroke. If left unaddressed, the consequent increased incidence of late arrival or refusal to seek care, limited treatment options and patient discontinuation of treatment will negatively affect optimal patient outcome.

Second, the evidence of sociocultural or religious beliefs and practices as a barrier to optimal care also deserves attention, particularly since this has received little attention in the current literature . More research in other settings may be useful to unravel the extent to which such practices influence provision of optimal care. In addition, national and local public awareness campaigns to increase the health literacy levels of the populace regarding stroke risk factors, early stroke symptoms and the need to seek early medical care are critical. This level of public awareness and education campaigns have been implemented in high-income countries such as UK,[61 62] Australia,[63 64] Canada[65] and Germany.[66] Hence, Ghana and other low/middle-income countries can clearly draw lessons from such public awareness campaigns on early recognition of stroke symptoms to minimise patient delays to seek care. Such interventions however would need to be adapted to suit the particular country and health context. They should be mainstreamed in the healthcare systems of such countries through collaboration with the public and private sectors to optimise the impact.

Importantly, the finding of patient discharge against medical advice has revived debates about the place of patients within the current evidence-based medicine paradigm where patients' needs and preferences are essential.[67] While more research on the implications of this issue is required, institutional measures exploring safe and appropriate times and conditions under which such requests could be granted should be identified. More importantly, strategies need to be adopted to ensure that requests by patients and families to be discharged against medical advice be counterbalanced with tailored communication and public campaigns to improve awareness of the risks and benefits. The roles of clinical psychologists and nurses can be pivotal in such communication. This has the potential to minimise the incidence of patient discharge against medical advice.

The limited collaboration and poor communication among stroke care professionals also warrants attention. Highlighted as an imperative in providing optimal acute stroke care by previous research,[12 54 55] the finding in this study further emphasises the need to explore effective ways to build collaborative working environments. Structural policy reforms are needed to ensure equal respect for individual professional experiences, identity, autonomy and responsibilities. This may be in the form of healthcare professional trainings, educational meetings and conferences, workshops to explore ways of improving clinical outcomes. As indicated earlier, staff professional development plays a critical role to stroke care quality improvement[68] and overall health outcomes,[69] and as such, efforts to provide staff educational and professional development opportunities in stroke care could be useful in the Ghanaian setting.

Given that health system and hospital level factors were observed as important to stroke care, strengthening health systems through the provision of adequate and effective acute stroke care services is essential. For example, to address the issue of limited staff numbers, an immediate short-term measure would be to consider task shifting approaches, as has been trialled in Nigeria. This Nigerian study[70] showed improved knowledge of non-neurologists in acute stroke care, thus potentially translating into improved patient outcomes.

Another health system barrier which has critical implications is the reported arrangement of upfront payment by patients prior to delivery of healthcare services . This suggests that people with symptoms of stroke and other emergency conditions such as heart attack and asthma may be provided optimal care on condition the patient is able to pay for such services. A health policy effort to expand the current package of the Ghanaian health insurance policy to cover the cost of CT brain scanning services will be in the right direction. In line with this, regular reimbursement of claims by the appropriate state institutions may address the issue of upfront payment prior to care.

Finally, to increase implementation of evidence-based acute stroke care, there is the need for increased policy commitment for optimal acute stroke care through increased allocation of resources to hospitals in the form of infrastructural support, a comprehensive coverage of the current national health insurance policy to include CT brain scan services and medical expenses for chronic care, staff professional development opportunities, and development of a stroke-specific clinical guideline are urgently needed.

## LIMITATIONS AND STRENGTHS

As a limitation, this study reported results from a limited set of participants whose views may not be reflective of the wider health staff responsible for acute stroke care in Ghana. Nonetheless, this was conducted in six major referral hospitals in 6 of the 10 administrative regions of Ghana and so the findings may be applicable to other stroke care professionals. Future studies should target a larger and more representative study sample to also include health planners and administrators in district and municipal hospitals. Further, given the qualitative nature of the study, the use of a semistructured interview guide and data interpretation could be subjective and thus, caution should be applied in interpretation. Nevertheless, using a robust reporting guideline, participant crosschecking and validation of interview transcripts, and the consistent use of the interview guide during the interview process minimised any possibility of bias but rather enhanced the study validity and reliability. Another key

limitation is the lack of observational or documentary evidence which could have accounted for any potential important information which may not have been shared by the participants during the interview process.

Notwithstanding these limitations, this study also has several strengths. First, to our knowledge, this study represents the first in Ghana to explore, in-depth, the barriers perceived by stroke care professionals to optimum provision of acute care in hospital settings. Another key strength is its exclusive focus on the perspectives of acute stroke care professionals from diverse professional disciplines, expertise, gender and tertiary and non-tertiary hospitals across different geographical settings. Using qualitative design, the findings provide contextually rich information of such barriers which would have been more difficult to unravel quantitatively. The study findings provide some new insights of other factors which have been less recognised in previous literature on the barriers to evidence-based acute stroke care. Added to this, this study did not focus on barriers to a specific stroke care intervention, as reported extensively in previous works, but rather on barriers across the continuum of stroke care.

## CONCLUSION

Overall, the views on barriers to optimal stroke care varied significantly based on specific professional discipline and study sites. Although most of the barriers were largely consistent with previous studies in high-income countries, the study unravelled some unique barriers which extend the body of literature on barriers to acute stroke care. Importantly, barriers in low/middle-income countries showed important differences to those from high-income countries. Greater political will for acute stroke care in terms of increased coverage of the national health insurance scheme, increased resource allocation, recruitment and training of an expanded stroke health workforce could improve uptake of evidence-based acute stroke care interventions. The information provided in this paper is potentially important to health managers, policy-makers, patients, grant managers or holders and other health stakeholders as it presents various reasons why delivery of acute stroke care in clinical setting may be far from optimum. To this end, to translate current evidence-based acute stroke interventions for optimal patient outcomes in Ghana and potentially in other resource-poor settings, a clear-cut understanding of these barriers to inform policy formulation, quality improvement and staff professional development, is critical.

**Acknowledgements** The authors thank the staff of the study hospitals for their support during field data collection. The authors also thank Dr Michael Otim, associate professor Liz McInnes and professor Sandy Middleton for their contribution to the study conception.

**Contributors** Study design and instrumentation process: LB, Ad-GA, GM, SS. Participants recruitment: LB, Ad-GA. Data collection, analysis, interpretation and writing of first manuscript: LB. Contribution to study interpretation and critical review of the manuscript: LB, Ad-GA, AS, GM, CKYC and SS. All authors have made substantial contribution to the writing of this manuscript for critical intellectual content. All authors have reviewed and approved the final version for submission.

**Funding** LB is a PhD candidate funded under the University International Students Scholarship programme. LB also received funding support from the University Faculty of Health Science Higher Degree Research (HDR) Student Support Scheme during his candidature. However, these funding bodies did not play a role in the study design, data collection and analysis, results interpretation, writing of the entire manuscript and the decision to submit the manuscript to this journal.

**Competing interests** None declared.

**Ethics approval** Australian Catholic University Human Research Ethics Committee (2015-154H), the Ghana Health Service Ethical Review Committee on Research Involving Human Subjects (GHS-ERC: 11/07/15), the Committee on Human Research Publications and Ethics of the School of Medical Sciences of the Kwame Nkrumah University of Science and Technology and the Komfo Anokye Teaching Hospital (CHRPE/AP/141/16) and lastly the institutional Review Board of the 37 Military Hospital (37MH-IRB IPN 035/2015).

**Provenance and peer review** Not commissioned; externally peer reviewed.

**Data sharing statement** No additional data are available.

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
