## [Reviewer comments · BMJ Open]

ARTICLE DETAILS

TITLE (PROVISIONAL)	Barriers to evidence-based acute stroke care in Ghana: A qualitative study on the perspectives of stroke care professionals
AUTHORS	Baatiema, Leonard; De-Graft Aikins, Ama; Sav, Adem; Mnatzaganian, George; Chan, Carina; Somerset, Shawn

VERSION 1 - REVIEW

REVIEWER	Rebecca Fisher University of Nottingham, UK.
REVIEW RETURNED	12-Dec-2016

GENERAL COMMENTS	This study raises some important issues associated with provision of acute stroke care in Ghana. On the whole the study was well designed and reported; I do however, have some comments requiring amendments, that I believe will strengthen the paper. Critically the key message for me was that there are some basic (and in some cases unique) issues relating to provision of care to stroke survivors in Ghana that need to be addressed urgently - this can then pave the way to address more detailed issues relating to working towards that care being fully evidence based. Overall focus on the study The authors talk about 'major advances in acute stroke care interventions' in relation to provision of 'optimal care' in the abstract objective section. Both phrases need more definition throughout the paper - particularly given the paper's title 'Barriers to evidence-based acute stroke care'. Are authors referring instead to 'major advances in acute stroke care research' - as this would fit better with the background section's focus on evidence based stroke care (and a research/clinical practice gap). I also think it needs to be clearer what the authors' definition is of 'optimal care' was - there's seems to be an undeclared assumption by the authors that 'optimal' is interchangeable with 'evidence based' - this needs to be clarified, particularly given the findings. Much of the data seems to relate to (a) general provision of acute care to stroke survivors in Ghana (i.e. a description of what is being provided - which remains an important exercise) - rather than (b) whether that care was evidence based; the authors need to be much clearer what they mean by optimal and evidence based, to fully address (b). I am also concerned that there seems to be a pre-occupation with thrombolysis - when there are many other features of evidence based acute stroke care to consider. The background section lacks
---

important references relating to core components of evidence based stroke care (e.g. Langhorne) and literature relating to provision (or lack of) of stroke rehabilitation (De Wit/Skarin).

Abstract objective and background sections need some revision based on comments above.

Abstract & Findings

The abstract results section could be worded better. Rather than 'four key barriers' which are then not specified in the abstract, I believe the data are structured relating to four levels (coding categories - patient factors, health system factors, healthcare provider factors and national policy context factors) - as referred to in table 2. Sub-themes could be grouped in relation to these four levels.

Methods/settings

The choice of the six out of ten administrative regions in Ghana needs more substantiation and the total number of hospitals, corresponding region and description (number of beds and admissions) needs to be clarified in the text and/or table 1.

Lack of specific protocol for acute care

From the information provided I wasn't convinced by this sub-theme and having looked at the interview schedule I would be concerned this was generated directly from the leading question (d) about whether guidelines or protocols were used, rather than from the data itself (a more open question would have been to ask what guided provision of care). This needs more substantiation.

The quotes themselves seem to relate more to nurse knowledge - which fits better with later sub-themes under healthcare provider factors.

Discussion

The authors refer to 'barriers found in high income countries' with the key point that they have identified unique barriers to optimal stroke care in low-middle income settings.

If a comparison is to be made, then the authors need to be clearer in the paper what barriers (specifically to provision of acute stroke care) have already been reported in high income countries. As in the background section, based on the references cited, there again appears to focus on thrombolysis, rather than wider literature relating to provision of evidence based acute stroke care.

In paragraph two, 'patient-level barriers' 'in line with previous research' - needs references.

Authors claim the 'low level of political will' for optimal acute stroke care 'seems unique to Ghana' - they need to substantiate this claim - I'm also not sure how this fits with the then acknowledged low prioritization of acute stroke care in resource poor settings.

Implications for future research

The authors need to be careful that this section doesn't read like an editorial piece - new information (e.g. cost of treatment in Australia) not directly relevant to the findings presented should be avoided. I think this section could be shortened to focus on implications directly relevant to findings featured in the discussion.

	Limitations and Strengths The authors should discuss the use of semi-structured interviews and potential bias - some of the questions were leading (is the provision of care guided by clinical guidelines or protocols?). Also what about information not readily articulated by stakeholders (observation or documentary evidence could have been used). Thank you to the authors for reporting findings of an important study.
--	--

REVIEWER	Peter Langhorne University of Glasgow, UK
REVIEW RETURNED	19-Dec-2016

GENERAL COMMENTS	I read this article with interest as it addresses the important topic of barriers to evidence based stroke care in a low and middle income country. In general, I found the article to be nicely conducted and clearly reported. I have only a few comments. 1 Healthcare planners and commissioners – was there a reason that you did not include healthcare planners and commissioners within your sample? This may be very important given the crucial role of funding and funding of service delivery. 2 Research implications – you have alluded to some research implications but can you comment on specific projects that could be carried out to tackle some of these identified barriers? 3 Discussion – the discussion refers to Australia as an upper middle income country. I believe this is incorrect, it should be classified as a high income country. I hope these comments are useful.
--

REVIEWER	William Meurer Department of Emergency Medicine University of Michigan, Ann Arbor, Michigan United States of America
REVIEW RETURNED	16-Jan-2017

GENERAL COMMENTS	The methods for this investigation are very well described and in accordance with recommendations for qualitative studies. While there are no major or critical flaws with this paper as written, I believe two areas could help readers across the world better contextualize the results. First, the term "acute stroke care" often refers to the very early period (within 6 hours of onset) and is focused on delivery of reperfusion in most areas of the developed world. Since you are using in a bit more broadly to also include the initial "acute care" hospitalization, where other aspects (nutrition and rehabilitation) are important, it might be useful to draw the distinction and make clear to the reader that you are focused on the initial days to weeks after a stroke, and not the initial hours. Second, in order to provide context, it may be helpful to organize the barriers into "general features of the Ghanaian health care system" (like needing to pay for medications up front) as opposed to "specific
--

	barriers that have Ghana/sub Saharan implications for stroke care" (such as the cultural beliefs regarding a stroke diagnosis and resistance to it). A paragraph of the discussion could be added to elaborate on the implications of the "health-system" (affects all sudden, unpredictable health conditions such as injury, infectious disease, stroke, etc) versus the "stroke-specific" barriers identified in your inquiry.
--	---

VERSION 1 – AUTHOR RESPONSE

	Reviewers' Comments	Response
Reviewer 1	Overall focus on the study	
	The authors talk about 'major advances in acute stroke care interventions' in relation to provision of 'optimal care' in the abstract objective section. Both phrases need more definition throughout the paper - particularly given the papers title 'Barriers to evidence-based acute stroke care'.	We have now provided a definition of evidence-based acute stroke care which has been used interchangeably with the term optimal stroke care throughout in this article. Please see lines 98-101.
	Are authors referring instead to 'major advances in acute stroke care research' - as this would fit better with the background section's focus on evidence based stroke care (and a research/clinical practice gap)? I also think it needs to be clearer what the authors' definition is of 'optimal care' was - there's seems to be an undeclared assumption by the authors that 'optimal' is interchangeable with 'evidence based' - this need to be clarified, particularly given the findings.	Thanks for this suggestion. We have now replaced it with the phrase "major advances in acute stroke care research". Please see line 32. As clarified earlier, optimal acute stroke care has been used interchangeably with the term evidence-based acute stroke care in this paper. An operational definition of optimal stroke care has now been added. Please see lines 98-101.
	Much of the data seems to relate to (a) general provision of acute care to stroke survivors in Ghana (i.e. a description of what is being provided - which remains an important exercise) - rather than (b) whether that care was evidence based; the authors need to be much clearer what they mean by optimal and evidence based, to fully address (b).	We agree with the Reviewer's observation that much of the data relate to general provision of acute stroke care to stroke survivors. However, evidence of lack of thrombolysis in all the study sites and the availability of a stroke unit care in only one study site clearly reflect evidence of the limited nature of some evidence-based acute stroke care interventions. The factors accounting for such limitations shed light on for example, why there is limited stroke unit care or lack of thrombolysis for acute stroke care in all study sites. We have also clarified what we mean by evidence-based acute stroke care in this paper as noted previously.
	I am also concerned that there seems to be a pre-occupation with thrombolysis - when there are many other features of evidence based	We agree with the Reviewer's observation about the emphasis in our paper towards thrombolysis. This was due to the fact that, to date, studies on barriers to the

acute stroke care to consider. The background section lacks important references relating to core components of evidence based stroke care (e.g. Langhorne) and literature relating to provision (or lack of) of stroke rehabilitation (De Wit/Skarin).	provision of acute stroke care have also focused more on thrombolysis. Nevertheless, we have now revised the background section to include other relevant literature on some other components of acute stroke care.
Abstract & Findings	
The abstract results section could be worded better. Rather than 'four key barriers' which are then not specified in the abstract, I believe the data are structured relating to four levels (coding categories - patient factors, health system factors, healthcare provider factors and national policy context factors) - as referred to in table 2. Sub-themes could be grouped in relation to these four levels.	We thank the Reviewer for this suggestion. As seen in the abstract section, the results have been regrouped according to these four key barriers. Please see lines 48-57.
Methods/settings	
The choice of the six out of ten administrative regions in Ghana needs more substantiation and the total number of hospitals, corresponding region and description (number of beds and admissions) needs to be clarified in the text and/or table 1.	We have made revisions to address this recommendation. However, incorporation of additional information on the study hospitals in Table 1 may compromise readability in our view. A supplementary file 2 has been provided to clarify this. Thank you.
Lack of specific protocol for acute care	
From the information provided I wasn't convinced by this sub-theme and having looked at the interview schedule I would be concerned this was generated directly from the leading question (d) about whether guidelines or protocols were used, rather than from the data itself (a more open question would have been to ask what guided provision of care). This needs more substantiation. The quotes themselves seem to relate more to nurse knowledge - which fits better with later sub-themes under healthcare provider factors	Thanks for the very useful observation. We agree with the Reviewer that this specific question as currently stated in the interview guide is potentially leading and not particularly open. However, there seems to be an error in how the interview guide was framed. The actual question that was presented to participants was a little different from what was shown in the earlier guide. We apologise for this error. The actual question presented to participants relates more to what guided the provision of care as per your suggestion. That question has thus been revised. See supplementary file 3. Regarding the presentation of the quotes, the general tone and direction of the discussion with participants during the interview process relates this sub-thematic barrier (lack of specific stroke care protocol) more to the responsibility of the health planners and managers at the organisational level.

		Further, we agree with your suggestion that the quotes related to the use of protocols could fit better with the health professional subtheme of barriers. However, the general direction of the discussion was the lack of protocol to support the provision of acute stroke care and not the lack of knowledge on how to use it. In our view, also, the availability of guidelines or protocols and other infrastructure are issues that can only be provided by the health care organisation and this explains why we allocated this subtheme of barrier at the health system level and not at the level of the health care providers.
	Discussion	
	The authors refer to 'barriers found in high income countries' with the key point that they have identified unique barriers to optimal stroke care in low-middle income settings. If a comparison is to be made, then the authors need to be clearer in the paper what barriers (specifically to provision of acute stroke care) have already been reported in high income countries. As in the background section, based on the references cited, there again appears to focus on thrombolysis, rather than wider literature relating to provision of evidence based acute stroke care. In paragraph two, 'patient-level barriers' 'in line with previous research' - needs references. Authors claim the 'low level of political will' for optimal acute stroke care 'seems unique to Ghana' - they need to substantiate this claim - I'm also not sure how this fits with the then acknowledged low prioritization of acute stroke care in resource poor settings.	As suggested we have now clarified this by reporting some specific barriers which were reported in developed countries in comparison to low-middle income countries. See lines 437 - 442. We agree with the Reviewer that there is more emphasis on barriers relating to the provision of thrombolysis. This is indeed the actual unbalanced picture of the literature which to date has focused more on the barriers associated with the provision of thrombolysis rather than on the overall barriers to provision of evidence-based acute stroke care. We have now included relevant references to this section as recommended. Please see line 459. Thank you. We agree with you on this suggestion. We have made some modifications to that sentence to clarify our point. Please see line 499.
	Implications for future research	

	The authors need to be careful that this section doesn't read like an editorial piece - new information (e.g. cost of treatment in Australia) not directly relevant to the findings presented should be avoided. I think this section could be shortened to focus on implications directly relevant to findings featured in the discussion.	Thanks for this observation. We have made some revisions to remove any information not directly relevant to our findings. Please see the changes introduced from 516-519. We have also made some deletions which have shortened the section under implications for research, policy and clinical practice. See lines 516-519, 571-574
	Limitations and Strengths	
	The authors should discuss the use of semi-structured interviews and potential bias - some of the questions were leading (is the provision of care guided by clinical guidelines or protocols?). Also what about information not readily articulated by stakeholders (observation or documentary evidence could have been used).	Thank you for this suggestion. We agree with the Reviewer and as suggested we have now incorporated them as part of the study limitations. Please see lines 581-588. As indicated earlier, we have clarified the comment on the wording of leading question which we hope helps address this important issue you noted.
Reviewer 2		
	Healthcare planners and commissioners – was there a reason that you did not include healthcare planners and commissioners within your sample? This may be very important given the crucial role of funding and funding of service delivery.	Thank you for this important suggestion. We agree that including health planners and commissioners would have been an important addition to the study participants but our study focused exclusively on acute stroke care professionals. Based on this query, this has been recommended in future research in this area. This can be found in line 580.
	Research implications	
	You have alluded to some research implications but can you comment on specific projects that could be carried out to tackle some of these identified barriers?	We welcome this recommendation. We have now provided some specific projects that could be explored to address the identified barriers. See lines 523-524, 539, 546-548, 562-565. Thank you
	Discussion	
	The discussion refers to Australia as an upper middle income country. I believe this is incorrect, it should be classified as a high income country.	Thanks for this correction. Yes we agree this is incorrect. We have now corrected this as seen in lines 516-518.
Reviewer 3		
	First, the term "acute stroke care" often refers to the very early period (within 6 hours of onset) and is focused on delivery of reperfusion in most areas of the developed world. Since you are using in a bit more broadly to also include the	We welcome this suggestion. This has now been clarified. See lines 127- 128. Thank you.

	initial "acute care" hospitalization, where other aspects (nutrition and rehabilitation) are important, it might be useful to draw the distinction and make clear to the reader that you are focused on the initial days to weeks after a stroke, and not the initial hours.	
	Second, in order to provide context, it may be helpful to organize the barriers into "general features of the Ghanaian health care system" (like needing to pay for medications up front) as opposed to "specific barriers that have Ghana/sub Saharan implications for stroke care" (such as the cultural beliefs regarding a stroke diagnosis and resistance to it). A paragraph of the discussion could be added to elaborate on the implications of the "health-system" (affects all sudden, unpredictable health conditions such as injury, infectious disease, stroke, etc.) versus the "stroke-specific" barriers identified in your inquiry.	We thank the Reviewer for this suggestion. We considered remodelling the discussion by reorganizing the order of the barriers based on your suggestion. However, the present organisation of the discussion better reflects how the data emerged from our interviews. We would therefore prefer to maintain the present structure. As suggested we have elaborated on the implications of the health systems related barriers. Please see lines 558-565.

VERSION 2 – REVIEW

REVIEWER	Peter Langhorne Glasgow University, Glasgow, UK
REVIEW RETURNED	01-Mar-2017

GENERAL COMMENTS	Thank you for addressing my concerns. There are typographical errors on: Line 153 Research Team Line 217 Line 500 Line 681
--

REVIEWER	William J. Meurer University of Michigan, USA
REVIEW RETURNED	17-Mar-2017

GENERAL COMMENTS	The authors have nicely addressed all of my concerns. I am supportive of this manuscript in the current form and have no residual concerns.
--